# OpenReview forum: "Position: Retrieval-augmented systems can be dangerous medical communicators"
_ICML.cc/2025/Position_Paper_Track — ICML 2025 Position Paper Track poster_

### Official Review · Reviewer_nZ3v · 2025-02-17

**Significance:** 3
**Argument Clarity:** 3
**Rating:** 3
**Confidence:** 5

**Questions:**

1: Why are medical-specific LLM e.g. OpenBio, MedLlama not being used in this study?

**Discussion Potential:**

3

**Paper Summary:**

Through a large-scale analysis of queries on disputed diagnoses and procedure safety, the authors present evidence that current AI systems decontextualize facts, omit critical sources, and reinforce patient misconceptions. Unlike human physicians, RAG-based models interpret queries narrowly and literally, failing to consider pragmatic communication and the broader context of medical advice. This can lead to dangerously misleading interpretations that appear well-grounded but lack an understanding of real-world consequences.

To mitigate these risks, the paper proposes improving AI systems with pragmatic reasoning, enhanced comprehension of source materials, and a better ability to recognize and address patient biases. These recommendations extend beyond medicine, underscoring the need for AI-generated information to align more closely with how human experts communicate. The study highlights that current RAG-based medical AI, despite citing reliable sources, often functions as a misleading and potentially harmful medical communicator.

**Position:**

Yes

**Position In Title:**

Yes

**Related Work:**

2

**Strengths And Weaknesses:**

Strength:
+ Give a good interpretation and hypothesis on the possible rationals why RAG systems tend to give narrow and literal interpreters of patient intent
+ The author points out the right way in healthcare how the philosophy of a proper health-RAG should be: "attentive and empathetic human expert. Rather than leaving important consequences of communicative reasoning implicit within systems trained towards other, more distant objectives – leading to brittle and often deeply undesirable results"

Weaknesses:
- Failure of RAG is mainly due to a systematic problem of training datasets, alignment, RAG system design (Graph-based vector-based may reflect huge performance differences), this paper mainly discusses the points with regards to limited decontextualising facts, and not reasoning about the downstream implications and consequences, this is not a comprehensive conclusion.

- The solution to improve the existing RAG system is very broad, e.g. the author mentioned about

"These structured representations might provide the basis for more sophisticated reasoning about patients’ queries or even repeated strings of queries, likeultimately inferring potentially unknown but important diagnoses with respect to repeated queries about symptoms
which stem from likely underlying causes. "

This lack of technical explanation of how this will be achieved, in the current scheme of training the LLM, it is unlikely to design a general loss function which can drive a representation space has reasoning while also maintaining relevancy toleratorance and less hallucination-prone to the rare and unknown space.

A more technical tangible direction should be discussed in this work.

- Qualitative results are presented, but lack quantitative ones

**Support:**

2

---

> ### Author Rebuttal · Authors · 2025-04-01
>
> Thank you for the thoughtful response. To respond directly to the specific comments:
>
> _“this paper mainly discusses the points with regards to limited decontextualising facts, and not reasoning about the downstream implications and consequences”_: our goal here is to argue for a more precise overarching framework that characterizes a range of intuitively undesirable behaviors. We argue directly that **failure to reason about pragmatic implications** (indeed, _specifically_ about the downstream consequences on a user’s beliefs and actions) is a unifying source of current failures, and should be the central focus of future RAG systems.
>
> In S2/S3, we use fact decontextualization as a running example of this more general point for two reasons: (1) it exemplifies the difference between literal accuracy and pragmatic implications, because decontextualized facts are _not_ hallucinations (they can literally reproduce text in the original source), yet yield undesirable pragmatic implications; and (2) it offers a concrete basis for measurable evaluations like in S2.
>
> However, the broader discussion of downstream implications is the unifying focus of the examples we provide throughout our paper.
> For instance:
> - The discussion of _omissions_, as with our other examples, centers on downstream consequences. We discuss queries about safety and complications precisely because these omissions have implications for patient beliefs / mental states (e.g., health anxiety), which in turn impacts patient actions (e.g, whether they decide to proceed with the procedure.) However, **see our response to reviewer NDfqb:** we will also update our manuscript with additional physician-provided examples of real world consequences.
> - More generally, each example in Fig. 2 (and throughout S3) is chosen to exemplify undesirable effects on patient beliefs and actions due to pragmatic miscommunication; in Fig. 2 alone, these include misinformation about diagnoses, treatments, relevant information for surgical choices, and about disease risk factors. The camera ready can emphasize these consequences.
>
> _“This lack of technical explanation…is unlikely to design a general loss function.”_
> - Our broader point throughout is that current RAG-systems train on objectives designed to reward **literal factuality and relevance**, when we argue that they should instead focus on **pragmatic interpretation and consequences**. We will update the camera ready to explicitly mention this as a general objective that can influence design at many levels – test-time decoding, UI, and alignment objectives. This is a position paper, not a methods paper, and our principal position makes this case by analyzing existing systems.
> - However, our technical recommendations do largely focus on explicit Bayesian models and UI design. With the extra page afforded by the camera ready, we will update Section 4 with an additional discussion of **potential aligned loss functions and training objectives based around pragmatic implications**, as our central position can be used to design these as well. For instance:
>    - To address pragmatic implicature relative to sources: we can suggest pragmatic, simulated ‘listener-speaker’ objectives during training (based on the RSA framework in Goodman and Frank 2016) which might, for instance, adapt actor-critic-like training objectives to evaluate pragmatic recovery of the original source content. We might design objectives around how well a simulated reader can recover other aspects of original sources; or, similarly, how surprised they would be to encounter other content in the source document given the generation.
>     - To address longer term consequences – we already discuss the value of gathering extended dialogues (eg. with clinicians in EHRs). We will discuss how these could provide training supervision on the end epistemic states of human physician-patient interaction. A useful RL objective might simulate follow up queries and reward how well simulated patients recover information a human physician would have provided in follow ups.
>
> _"Qualitative results are presented, but lack quantitative ones"_: see response to reviewer NDfQ re: empirical human evaluation, but S2 does provide quantitative analyses in a large-scale query analysis.
>
> _“Why are medical-specific LLM…not used in this study?”_
>
> Two main reasons:
> - (1) We focus on consumer-facing RAG systems as these are the primary ones used by current patients;
> - (2) Those med-specific LLMs are not integrated into existing RAG systems for patient use. We wanted to focus on widely available (highly engineered, aligned) systems.
>
> However: in ongoing work, we also found that med-specific LLMs (e.g. OpenBioLLM, Meditron) are **largely as susceptible to presupposition in generating medical misinformation** as their base models (Med LLMs produced misinformation 27% of the time for presuppositional prompts, vs. 31% for base models). We will allude to this work in our camera-ready.

---

### Official Review · Reviewer_NDfQ · 2025-03-13

**Significance:** 2
**Argument Clarity:** 2
**Rating:** 3
**Confidence:** 3

**Questions:**

- Can the authors provide more details on how LLM was used as a judge to evaluate misleading responses? What are the criteria?
- How are the temporal variations in Google AI Overview results were accounted for?
- Is it possible to use human evaluation to assess whether users perceive responses as misleading or how they influence decision-making.
- Is Jaccard similarity enough for statistical analysis of response biases?

**Discussion Potential:**

3

**Paper Summary:**

This paper examines the risks of retrieval-augmented generation (RAG) systems in medical communication, arguing that even when these systems generate factually accurate responses, they can still be misleading due to their literal interpretation of patient queries, decontextualization of sources, and failure to consider downstream consequences. The study includes a large-scale query analysis of Google AI Overview and Perplexity AI, showing how these systems reinforce biases, omit critical context, and misrepresent medical information. The authors propose solutions for improving AI-generated medical communication by incorporating pragmatic reasoning about user intent, source credibility, and response implications.

**Position:**

Yes

**Position In Title:**

Yes

**Related Work:**

3

**Strengths And Weaknesses:**

**Strengths**
- This paper addresses an urgent issue in AI-driven medical information dissemination, given the increasing reliance on AI for health-related queries.
- The large-scale query analysis in this paper is comprehensive and provides insights on the potential causes of the failures.
- This paper also proposes actionable recommendations to improve the safety and effectiveness of RAG systems in the medical domain moving forward.

**Weaknesses**
- Some methodological clarifications are needed. Please see questions.
- There lacks of quantitative validation of user impact. It would be nice to add human evaluation to the experiment.
- This paper does not address the ethical concerns and potential bias in the experiment.

**Support:**

3

---

> ### Author Rebuttal · Authors · 2025-04-01
>
> Thanks for the careful review! We’re glad to hear that you agreed with the urgency of the issue we've raised and the actionability of our solutions. To address your specific questions:
>
> - _"Can the authors provide more details on how LLM was used as a judge to evaluate misleading responses? What are the criteria?"_
> Since we had a controlled experiment, we could evaluate misleading responses with very specific criteria. For disputed medical diagnoses, we used the LLM to judge whether each response mentioned at all that the condition is considered controversial. Concretely, the model was asked to “Determine whether the following passage mentions whether {topic} is pseudoscientific and controversial.” For surgery complications, we used the LLM to judge whether each response mentioned (i) any statistic mentioning the rarity of the complication or (ii) any benefit of receiving the surgery. All surgeries probed are standardly conducted with consensus on benefits. Exact prompts can be found in our supplementary materials in auto-annotation/evaluator.py. We used two versions of GPT-4o, spot-checked the labels for quality, and manually adjudicated responses when the two versions disagreed. We will add these details to the appendix.
> - _"How are the temporal variations in Google AI Overview results were accounted for?"_
> The Google results fluctuate, so we collected and averaged results over 3 consecutive days to achieve a more robust estimate. While answers differed, we did not identify any systemic difference across choice of dates.
> - _"This paper does not address the ethical concerns and potential bias in the experiment."_
> Since our quantitative results did not involve any human experiments, we didn’t have specific ethical concerns in the experimental design. However, we acknowledge that there could be some degree of bias in the choice of disputed medical diagnoses; see our response to reviewer eJxY for how these were sourced.
> - _"Is Jaccard similarity enough for statistical analysis of response biases?"_
> We use Jaccard similarity only to specifically analyze the similarity of cited downstream sources for queries with different framings. This analysis reveals that RAG-based systems provide significantly different sources, indicating a significant effect of confirmation bias. Other behaviors are investigated through other parts of the analysis.
> - _"Is it possible to use human evaluation to assess whether users perceive responses as misleading..."_
> Thanks for the suggestion, which we had also considered ourselves. Unfortunately, a human evaluation would be tricky, since it would involve exposing users/patients to potentially misleading information and seeing whether they believed it. As you can imagine, IRB’s are understandably cautious about any such study design that would purposefully expose patients to misinformation, due to the potential real-world ramifications. That being said, we will add to our paper related work that shows that the behavior we identify
> in RAG models are the same behaviors that lead to belief of misinformation in the existing cognitive science literature. For example, “The cognitive foundations of misinformation on science” summarizes how belief of misinformation stems from phenomena including trust of familiar sources, confirmation bias, and desire for simplicity.
> - _"Is it possible to use human evaluation to assess ... how they influence decision-making."_
> Despite the difficulty of controlled experiments, one of our co-authors is a physician, who has anecdotally noticed this influence on patient decision-making already. We provide thoughts from their perspective below, and we will summarize these in our camera-ready to ground the urgency of need: “Because
> LLMs tend to affirm questions and biases, for patients with fears or hesitations
> about interventions, LLMs can lead to disruption of a natural conversation
> between a physician and patient by providing patients with evidence for harm
> out of proportion to the benefit of an intervention. I’ve observed this with an
> acquaintance with a newly diagnosed condition. Over a phone call, they asked
> questions centered around risks with incidences of
> less than a fraction of a percent. They admitted to sourcing information from
> AI-generated text due to being nervous about the procedure. This resulted in
> extra work to re-direct the conversation to a typical starting point, where we
> could discuss the overall diagnosis, work-up, and possible outcomes. In another
> example, an acquaintance contacted me describing symptoms that necessitate
> further evaluation. Unfortunately, they had had the symptoms for weeks, but
> searching on the Internet (and specifically AI-generated text) reassured them that
> it was benign. RAG based LLMs do not ask for further information to develop
> differentials, which can be problematic and falsely reassuring. Prior to LLMs,
> online health information required reading primary sources.”

---

### Official Review · Reviewer_eJxY · 2025-03-13

**Significance:** 4
**Argument Clarity:** 3
**Rating:** 4
**Confidence:** 4

**Questions:**

1. Although I can understand the claims and the general logics, I am still confused what "disputed medical diagnoses" mean in this paper. The appendix provides a list of phrases but that does not seem to give more information.

**Discussion Potential:**

4

**Paper Summary:**

The paper studies the current limitation of RAG-based medical AI systems, using Google’s AI Overview (Google AIO) and Perplexity AI as the target systems. The paper did both quantitative (Sec 2) and qualitative analysis (Sec 3), as well as some suggestions (Sec 4).

In Section 2, it show that these systems often give correct facts, but the way they answer can still be misleading because they take the questions very literally. For example, if someone asks about a health condition that is controversial (meaning experts don't fully agree on it), the search engines' answers change a lot depending on how the question is phrased. Also, when asking about the safety of medical procedures, the search engines give very different answers based on whether the user asks if the procedure is safe or dangerous.

In Section 3, it delves into why responses generated by retrieval-augmented systems can be misleading compared to what a human clinician might provide. It emphasizes that these systems typically lack the pragmatic reasoning needed to interpret the real intent behind queries and the source content they use.

Section 4 outlines potential strategies for improving retrieval-augmented systems to serve as more effective and safe medical communicators. It suggests a comprehensive rethinking of how these systems handle the interpretation of user queries, source intentions, and the communicative goals of their responses. The paper advocates for systems that can more deeply understand and align with the informational and emotional needs of users.

## Update after rebuttal: The additional information provided by the authors is very helpful. I do not have big concerns on this work.

**Position:**

Yes

**Position In Title:**

Yes

**Related Work:**

3

**Strengths And Weaknesses:**

Strengths:
1. The paper discusses an interesting, crucial, and very practical aspect of LLM-based medical AI systems. I believe it will motivate better design of the future systems.
2. The paper includes both quantitative and qualitative analysis.
3. The proposed suggestions of future work are, in general, promising to me. It highlights the important of reasoning, such as "A patient already searching for the symptoms of a disease probably believes that the disease exists; a patient searching for why a treatment is controversial likely has both a particular prior about the treatment and hopes to use that information for some downstream goal".

Weakness:
I did not find significant weakness of this paper. Some parts can still be improved. At the beginning (including Sec 2), the paper is difficult to read, perhaps because the ideas are expressed in a very high-level and abstract way without concrete examples.

**Support:**

3

---

> ### Author Rebuttal · Authors · 2025-04-01
>
> Thank you for the thoughtful review! We’re glad to hear that you resonated with the core points. We’re particularly excited to be focusing on this position as well, as we see pragmatic interpretation (and the potential for misinterpretation) as particularly important for medical language models.
>
> To address some of the specific questions and comments:
>
> - _“At the beginning (including Sec 2), the paper is difficult to read, perhaps because the ideas are expressed in a very high-level and abstract way without concrete examples.”_: this is useful to hear. With the extra page granted for a camera-ready, we’ll update the paper in revision to include more concrete examples earlier on in the introduction (and can also bring up some of the examples from Figure 2 and the later sections earlier on in the text.)
>
> - _“Although I can understand the claims and the general logics, I am still confused what "disputed medical diagnoses" mean in this paper. The appendix provides a list of phrases but that does not seem to give more information.”_: Here’s a definition of disputed medical diagnoses given by the physician author of this paper: “Disputed medical diagnoses refer to diagnoses that do not have consensus on definition, pathophysiology, treatment, or even existence. For example, hypertension is not a disputed diagnosis, but chiropractic spinal subluxation is, as the clinical consensus around the majority of medical doctors would be to cite a lack of evidence to support its pathophysiology or effectiveness of treatment.” More concretely, the specific disputed medical diagnoses used in this paper were sourced directly from the Wikipedia list of “diagnoses characterized as pseudoscience” (https://en.wikipedia.org/wiki/List_of_diagnoses_characterized_as_pseudoscience), but these were additionally reviewed by the physician author on this paper and we removed a few syndromes (eg. Chronic Lyme disease and Leaky gut syndrome) that are better characterized by the medical consensus as more active areas of research, rather than directly pseudoscientific in nature. We will update the camera ready to include both the source and this definition.

---

> > ### Comment · Reviewer_eJxY · 2025-04-04
> >
> > Thanks for the detailed explanation. That is very helpful. I do not have big concerns on this work.

---

### Decision · Program_Chairs · 2025-04-29

**Decision:**

Accept (poster)

**Comment:**

The reviewers agree that this paper's position is clear and well argued and thus would be a welcome addition to the program.  However, the reviewers were not uniformly positive.  While the identification of the underlying problem is clear, the paper falters in identifying how to address the problem and in grounding the problem with accessible examples.  I suspect that these could be addressed relatively easily in the camera ready.